# An International Consensus on the Design of Prospective Clinical–Translational Trials in Spatially Fractionated Radiation Therapy for Advanced Gynecologic Cancer

**DOI:** 10.3390/cancers14174267

**Published:** 2022-08-31

**Authors:** Beatriz E. Amendola, Anand Mahadevan, Jesus Manuel Blanco Suarez, Robert J. Griffin, Xiaodong Wu, Naipy C. Perez, Daniel S. Hippe, Charles B. Simone, Majid Mohiuddin, Mohammed Mohiuddin, James W. Snider, Hualin Zhang, Quynh-Thu Le, Nina A. Mayr

**Affiliations:** 1Innovative Cancer Institute, South Miami, FL 33143, USA; 2Department of Radiation Oncology, New York University, Langone Health, New York, NY 10016, USA; 3Department of Radiation Oncology, Dr. Negrin University Hospital, 35010 Las Palmas de Gran Canaria, Spain; 4Department of Radiation Oncology, University of Arkansas for Medical Sciences, Little Rock, AR 72205, USA; 5Executive Medical Physics Associates, Miami, FL 33179, USA; 6Clinical Research Division, Fred Hutchinson Cancer Center, Seattle, WA 98109, USA; 7Department of Radiation Oncology, New York Proton Center, New York, NY 10035, USA; 8Radiation Oncology Consultants and Northwestern Proton Center, Warrenville, IL 60555, USA; 9Radiation Oncology, Scottsdale, AZ 85258, USA; 10Department of Radiation Oncology, University of Alabama at Birmingham School of Medicine, Birmingham, AL 35233, USA; 11Department of Radiation Oncology, University of Southern California, Los Angeles, CA 90033, USA; 12Department of Radiation Oncology, Stanford University, Stanford, CA 94305, USA; 13College of Human Medicine, Michigan State University, East Lansing, MI 48824, USA

**Keywords:** spatially fractionated radiation therapy, Lattice therapy, GRID therapy, dose fractionation, radiation therapy, clinical trials, consensus guideline, gynecologic cancer, cervix cancer

## Abstract

**Simple Summary:**

Spatially fractionated radiation therapy (SFRT) delivers intentionally heterogenous dose to tumors. This is a major departure from current radiation therapy, which strives for uniform dose. Early pilot experience suggests promising treatment outcomes with SFRT in patients with challenging bulky tumors, including gynecologic cancer. Well-conducted prospective multi-institutional clinical trials are now needed to further test SFRT as a treatment modality. However, clinical trial development is hampered by the variabilities in SFRT approach and the overall unfamiliarity with heterogeneous dosing. A broad consensus among SFRT experts, potential investigators, and the wider radiation oncology community is needed so that clinical trials in SFRT can be successfully designed and carried out. We developed an international consensus guideline for the design parameters of clinical/translational trials in SFRT for gynecologic cancer. High-to-moderate consensus was achieved, and harmonized fundamental design parameters for SFRT trials in advanced gynecologic cancer were defined.

**Abstract:**

Despite the unexpectedly high tumor responses and limited treatment-related toxicities observed with SFRT, prospective multi-institutional clinical trials of SFRT are still lacking. High variability of SFRT technologies and methods, unfamiliar complex dose and prescription concepts for heterogeneous dose and uncertainty regarding systemic therapies present major obstacles towards clinical trial development. To address these challenges, the consensus guideline reported here aimed at facilitating trial development and feasibility through a priori harmonization of treatment approach and the full range of clinical trial design parameters for SFRT trials in gynecologic cancer. Gynecologic cancers were evaluated for the status of SFRT pilot experience. A multi-disciplinary SFRT expert panel for gynecologic cancer was established to develop the consensus through formal panel review/discussions, appropriateness rank voting and public comment solicitation/review. The trial design parameters included eligibility/exclusions, endpoints, SFRT technology/technique, dose/dosimetric parameters, systemic therapies, patient evaluations, and embedded translational science. Cervical cancer was determined as the most suitable gynecologic tumor for an SFRT trial. Consensus emphasized standardization of SFRT dosimetry/physics parameters, biologic dose modeling, and specimen collection for translational/biological endpoints, which may be uniquely feasible in cervical cancer. Incorporation of brachytherapy into the SFRT regimen requires additional pre-trial pilot investigations. Specific consensus recommendations are presented and discussed.

## 1. Introduction

Spatially fractionated radiation therapy (SFRT), the delivery of intentional highly *non*-uniform tumor dose distributions, presents a major departure from conventionally practiced radiation therapy which seeks the best possible dose homogeneity. SFRT has shown high, and sometimes drastic palliative tumor responses in challenging bulky treatment-refractory tumors, while causing only minimal toxicity [1,2,3,4,5,6,7,8,9,10,11,12,13]. Investigating the underlying mechanisms of SFRT’s profound tumoricidal effects is an area of active and rapidly advancing research. The combination of high ablative tumor doses in SFRT [4,14] with vast dose heterogeneity, which preserves intra-tumoral microenvironment and vasculature in low-dose regions (and thereby promotes immunomodulating and bystander effects [15,16,17,18]), are thought to contribute to the enhanced tumor response. Such underpinnings may be particularly promising in the currently increasing use of immune-modulating agents in radiation oncology [19].

Building on the encouraging experience in palliation [1,2,3,4], single-institution smaller pilot studies have since explored SFRT in primary (non-metastatic) advanced bulky primary head and neck [5,6,13], lung [20], cervical cancer [8], and sarcoma [9,10,12], treated with curative intent. Longer-term outcome data in these cohorts have shown similarly high responses to SFRT, with high local control and encouraging survival outcomes and limited toxicities in these far-advanced bulky primary tumors [5,6,7,8], as observed in the initial multi-disease palliative series [1,2,3,4]. The SFRT pilot experience in bulky advanced gynecologic cancers is promising [8,21,22]. Specifically, in SFRT for cervical cancer, high response rate and local control and encouraging survival have been observed [8,21] that mirror the results of the SFRT experience in other primary tumors [2,3,4].

However, no multi-institutional or prospective randomized clinical trials of SFRT have been conducted in gynecologic cancer. Such trials are important to rigorously evaluate the potential utility of SFRT in advanced stage, bulky gynecologic tumors. Gynecologic cancers present a continued therapeutic challenge, specifically cervical cancer. Advanced cervical cancer has a propensity to present as bulky/voluminous tumors [23], particularly in economically underserved communities and low-income countries [24]. Treatment outcomes are still unsatisfactory, and the recent disappointing results of adjuvant chemotherapy in stage IB (bulky)–IVA disease [25] have called the success of treatment intensification with systemic therapy into question.

The development of clinical trials in SFRT for gynecologic cancers is uniquely challenged by the non-intuitive dosing concepts of non-homogeneous irradiation that demands complex biological modeling; the overall unfamiliar dosimetric and physics metrics of SFRT; and the variable SFRT techniques and technologies/platforms, which can introduce inconsistencies into a clinical trial. In addition, dose heterogeneity patterns, which differ from those in preclinical SFRT studies, are variable. 

Consensus on a standardized approach of clinical trial design is needed regarding eligibility, SFRT dose prescription, dose reporting of the heterogeneity parameters, suitable endpoints, and the feasibility of translational trial components to elucidate the biological underpinnings of the SFRT effect, to enable clinical trials in primary gynecologic cancers, and to facilitate broad participation and successful accrual. 

To address challenges towards the development of prospective multi-institutional SFRT trials, the consensus development for clinical trial design of SFRT in primary malignancies has been a major effort of the *Radiosurgery Society (RSS) GRID*, *Lattice*, *Microbeam and Flash (GLMF) Radiotherapy Working Groups* [26,27]. In this initiative, major disease sites have been [28] and are currently evaluated for trial feasibility and clinical trial design consensus. Here, we report the clinical trial design consensus for gynecologic cancers. 

The purpose of this consensus guideline effort was to develop a common understanding and approach for the design of future prospective multi-institutional clinical trials of SFRT in gynecologic cancers, informed by the collective disease-specific clinical experience and by physics and biology considerations. Specifically, design criteria guidelines for SFRT clinical trials included eligibility, stratification, prescription dose/fractionation, target and normal tissue dose parameters for both the SFRT and conventional radiotherapy components of treatment, SFRT platform and technique, systemic therapies, pre-, on-, and post-treatment assessments for outcome endpoints, and the clinical feasibility of correlative science investigations. 

## 2. Materials and Methods

This consensus effort was carried out by the Clinical Working Group within the RSS GLMF Radiotherapy Working Groups. The GLMF Radiotherapy Working Groups had been established subsequent to a 2018 National Cancer Institute’s Radiation Research Program/RSS Workshop, and their goal is to advance the biological, technical/physical, and clinical understanding of these novel radiation approaches [26,27]. The consensus reported here is part of the GLMF Radiotherapy Working Groups project of developing design consensus for clinical SFRT trials in primary (non-metastatic) head and neck [28], lung, gynecologic cancer, and sarcoma [28].

The process of developing a clinical trial design consensus is summarized in Table 1 and has been previously described in detail [28]. In brief, the consensus followed a formal 12-step process as shown in the table. 

Initially a comprehensive search and *review of the SFRT literature*
*(step 1)* was performed with focus clinical studies that reported tumor control and toxicity outcomes of SFRT. Studies were reviewed and tabulated systematically according to parameters of study quality, technical parameters, and treatment outcome criteria. Literature evidence tables were constructed from this review. The literature evidence table for gynecologic cancers is presented in Appendix A. 

Based on pertinent clinical trial principles and the data from the literature review, *draft clinical trial design criteria (step 2)* were developed by a group of leading SFRT experts, and the most appropriate primary tumor sites to be considered for SFRT clinical trials were established. The overall draft criteria were then tailored according to gynecologic cancer. The domains of these draft clinical trial design criteria are presented in Table 2. A 19-question voting survey comprising 69 individual clinical trial parameters was developed. Tailoring of the general clinical trial criteria included specific consideration for SFRT techniques and technologies, dose-to-target coverage and normal tissues, brachytherapy, and considerations of combined modality radiation with chemotherapy and immunotherapy. 

*Consensus voting (step 3)* was performed through distribution of the voting survey of clinical trial design parameters and the literature evidence table (Appendix A) to national and international experts with publications, scientific presentations, and/or clinical SFRT practice in of SFRT in gynecologic cancers. Anonymous electronic rank voting (on a ranking scale of 1–9) was carried out by the experts. The voting process, *vote analysis and statistical model (step 4)* that was developed to interpret the rank voting data, have been described previously [28] and is presented in Table 1.

A disease-specific multi-disciplinary SFRT expert panel for gynecologic cancers (“panel”) was established based on expertise, and publication and presentation record on SFRT in gynecologic cancers. The aggregated rank voting results were shared among the panel. Consensus was carried out by the panel through iterative *review and discussion (step 5)* of the voting results, applying modified Delphi technique [29] principles. The voting results and assessments by the panel are presented in the consensus table (Appendix A). An *iterative voting round (*with *re-review of results, steps and 6 and 7)* were not required for gynecologic cancers per the panel’s assessment. 

The *draft consensus guideline for clinical trial design was developed*
*(step 8)* by the panel and was posted on the RSS website for *public comment (step 9*) for two weeks (from 26 April to 10 May 2022). The comments were reviewed by the panel, and after consideration of the comments, the guideline was finalized. 

## 3. Results: SFRT Clinical Trial Design Consensus Guideline for Primary Gynecologic Cancer

SFRT clinical trial design recommendations were informed and guided by studies of multiple disease sites that included gynecologic cancer patients [2,3,4], reports of individual gynecologic cancers [21,22], and by a *disease-specific* series of cervical cancer patients [8]. These data were considered in the consensus process in conjunction with the clinician, physicist, and biologist experience of the multidisciplinary expert panel for SFRT clinical trials in gynecologic cancer. Specific clinical trial criteria for gynecologic cancers are summarized in Table 2. The full consensus guideline can be accessed in Appendix A.

### 3.1. Eligibility 

Based on the patient characteristics of the published outcome studies [8,21], the panel determined that cervical cancer is currently the only primary gynecologic tumor site with sufficient pilot data to warrant an SFRT clinical trial. The few reports of SFRT in other primary gynecologic malignancies consist of patients with unspecified gynecologic primary tumors that are part of larger multi-disease series containing largely palliatively treated patients [2,3] and two ovarian cancer cases [4,22]. More pilot clinical outcome experience would be required to justify an SFRT trial in a gynecological cancer site other than cervical cancer. The consensus on eligibility and exclusion criteria is summarized in Table 3. Patients with advanced bulky squamous cell carcinoma, adenocarcinoma, and mixed adeno-squamous carcinoma of the cervix of FIGO stages IB2–IVA, with tumors ≥6 cm in largest diameter (by palpation and/or MR imaging), are eligible for trial enrollment (moderate consensus) based on their overall lower local control and survival outcomes. Inclusion of patients with tumors of at least 5 cm in largest diameter who meet strict criteria for ineligibility for brachytherapy (Table 3) can be considered for enrollment (moderate consensus). Patients with both uninvolved and involved pelvic or para-aortic lymph nodes are eligible. Less-common and biologically distinct histologies, including small cell neuroendocrine carcinoma, sarcoma, or lymphoma, are excluded (high consensus). 

Patients with recurrent tumors after prior hysterectomy or prior definitive radiation therapy for cervical cancer are ineligible for a trial of primary cervical cancer (high consensus). Surgical lymph node staging procedures are permitted. Patients with prior systemic therapy (chemotherapy or targeted therapy), such as neoadjuvant chemotherapy, are not eligible because of the confounding influence of pre-trial systemic therapy on the interpretation of outcome endpoints (high consensus).

Patients 18 years or older are eligible for enrollment, with no upper age limit as long as performance status is acceptable (high consensus). Patients at increased risk for normal tissue complications, such as a history of inflammatory bowel disease or scleroderma (systemic sclerosis), should be excluded (high consensus).

### 3.2. Stratifications 

The trial cohort should be stratified according to lymph node status (uninvolved vs. involved) due to the profound prognostic impact of lymph node status in this patient population. 

### 3.3. Endpoints 

The feasibility of delivering SFRT according to the dosimetric and physics specifications [14] (see section *Radiation Therapy: SFRT Dose*), and response metrics including tumor response and local control are suitable primary endpoints. Local progression-free, metastasis-free, and overall survival, toxicity, and quality of life outcomes present additional clinical trial endpoints. 

### 3.4. Radiation Therapy 

#### 3.4.1. SFRT Dose

Based upon outcome data in cervical cancer [8] and other gynecologic malignancies [22], as well as the multi-disease studies including gynecologic cancers [2,3,4], two dose regimens can be considered. The most studied schedule is 24 Gy in 3 fractions on consecutive days to the tumor target in the cervix. 

The dose schedule of 24 Gy in 3 fractions has been studied in 10 patients with highly advanced cervical cancer (stages IIIB-IVA with excessive tumor bulk of ≥7 cm and /or severe anatomical distortion) by Amendola et al. [8], who observed high clinical and molecular imaging response and very low toxicity (no grade >3 short or long-term toxicity). Among the cohort, five patients had adjuvant hysterectomy after high tumor response, and three of the five had no residual tumor in the surgical specimen (pathologic complete response) (personal communication, Dr. B. Amendola). No brachytherapy was given in this series. 

While the panel recognizes the schedule of 24 Gy in 3 fractions as the most-studied regimen, there was overall moderate consensus regarding the SFRT dose in cervical cancer. A single-fraction schedule has the potential advantage to avoid inter-fraction shifts of the high-dose peaks within the tumor, although such shift might not have the same implications as in the conventional RT. Conversely, a three-fraction schedule may be better-tolerated by critical normal tissues. While the corresponding isoeffective single-fraction regimen may be 16.5 Gy in 1 fraction (BED 43.7 Gy, EQD_2_ 36.4 Gy, α/β = 10 Gy, compared to BED 43.2 Gy, EQD_2_ 36.0 Gy, α/β = 10 Gy for 24 Gy in 3 fractions, applying radiobiologic modeling for uniform dose as an estimate), the clinical use of 16.5 Gy in 1 fraction has not been reported in cervical cancer, and the conventional BED or EUD formalisms have not been fully justified. 

A single fraction of 15 Gy Lattice radiation therapy (LRT) has been used in two cases (unpublished) as part of a recent regimen proposed by Larrea et al. [30], combined with full-dose conventional radiation with concurrent chemotherapy and brachytherapy. The panel, therefore, considers 15 Gy in a single fraction as a potential SFRT dose regimen in cervical cancer. However, the combination of this regimen with brachytherapy remains to be tested prospectively (see section *Radiation Therapy—Conventional Radiation Therapy: Dose and Technique: Brachytherapy*). 

Standardization of the SFRT prescription dose, defined as the peak dose, is mandatory. Dosimetric and geometric characteristics of the heterogeneous dose distribution, such as dose volume histogram parameters (e.g., D10, D50, D90), vertex diameter, vertex volume and vertex distance, valley dose, and peripheral target dose must be reported according to guidelines further described in the recent LRT physics and dosimetry white paper [31]. Robust dosimetry and dosimetry reporting are also important for advancing the understanding of tumor environmental and potential immune modulation effects of SFRT. 

The equivalent uniform dose (EUD) of the SFRT regimen must be determined for the trial regimen. This includes the EUD for cervical squamous cell and adenocarcinoma (using α/β = 10 Gy) and for critical normal tissues (generally α/β = 3 Gy). Current concepts favor the modified linear quadratic (MLQ) for EUD calculation because of its greater accuracy at doses of >10 Gy. However, the traditional formalism developed for whole target volume irradiation awaits justifications or modifications in the SFRT settings. Detailed models for EUD computation and qualifications/precautions are described in the recent SFRT physics guideline publications [14,31]. 

#### 3.4.2. SFRT Target Volume 

SFRT should be delivered to the primary (cervical) tumor. The GTV is the primary cervical tumor extent as defined by imaging. To account for organ motion, simulation with both full and empty bladder is recommended to establish an internal target volume (ITV). For the planning target volume (PTV), a 2–3 mm margin is added to the ITV because of the proximity of mobile sensitive normal tissue structurers, particularly small bowel, large bowel, and bladder (high consensus). For LRT, a Lattice volume (V_L_), which is the ITV minus an inward margin that allows for the dose to drop from the dose peaks (vertices) to the periphery, should be created. Vertices should be only placed within the volume that is common for both empty and full bladder scans. 

While SFRT is primarily given to the bulky primary tumor in the cervix, the delivery of SFRT to a bulky (≥6 cm) lymph node or lymph node conglomerate is permitted. In the case of SFRT to a bulky lymph node, the GTV includes the lymph node mass GTV plus a 2–3 mm margin to account for the proximity of mobile sensitive normal tissue structures (high consensus). 

#### 3.4.3. SFRT: Normal Organ-at-Risk Structures

For the three-fraction regimen of 24 Gy in 3 fractions, the dose to the periphery of the PTV should be limited to no more than 9 Gy in 3 fractions (BED 18 Gy, EQD_2_ 10.8 Gy for α/β = 3 Gy). For the one-fraction regimen of 16.5 Gy, the peripheral GTV dose is limited to no more than 6 Gy in 1 fraction (BED 18 Gy, EQD_2_ 10.8 Gy for α/β = 3 Gy) (high consensus). There is (unpublished) evidence that lower peripheral doses are achievable with LRT that maintain 2 cc doses to rectum, bladder, and sigmoid of <5–6 Gy (in 3 fractions) for a prescription dose of 24 Gy in 3 fractions. 

#### 3.4.4. SFRT: SFRT Technique 

LRT should be employed as the SFRT technique. In the absence of any studies using GRID therapy for the definitive treatment of cervical cancer, GRID therapy is not currently recommended for a clinical SFRT trial in cervical cancer (high consensus). For treatment delivery of the SFRT, it is emphasized that organ motion be managed, as further described in section *Radiation Therapy: SFRT Dose*, and daily image-guided therapy with pre-treatment cone beam CT is performed with stereotactic alignment (as is customary in stereotactic body radiation therapy). CBCT imaging is particularly important due to the frequently observed rapid tumor response that may require adaptive therapy using rapid replanning. In the series by Amendola et al. [8] adaptive therapy was needed in 7 of the 10 patients. 

#### 3.4.5. Conventional Radiation Therapy: Dose and Technique

*External beam radiation.* SFRT is followed by conventionally fractionated radiation therapy (cERT) to a dose of 45–50 Gy at 1.8–2 Gy per fraction to the whole pelvis and to tumor extension beyond the pelvis (e.g., para-aortic lymph nodes), as clinically indicated (high consensus). The use of IMRT to reduce normal tissue dose is highly encouraged (high consensus) in view of the proximity of sensitive normal tissues and the overall high cumulative target doses in cervical cancer, due to the use of brachytherapy, that tend to be higher than in other tumors treated with external radiation alone. In the SFRT literature for cervical cancer, the conventional doses to gross tumor PTV ranged from 39.6 to 45.0 Gy in 25 fractions Gy (combined with SFRT of 24 Gy in 3 fractions) [8]. 

Boosts to involved lymph node(s) can be delivered either sequentially (following whole pelvis radiation) or with a simultaneously integrated boost, as clinically indicated and under consideration of the OAR constraints described in section *Conventional ERT: OAR constraints*. An initial SFRT boost to a voluminous involved lymph node or matted lymph nodes is permitted, as described in section *SFRT Target volume* (high consensus). If brachytherapy is not an option, external beam as either conventionally fractionated IMRT under consideration of OAR dose limits or stereotactic hypofractionated boost to the residual tumor has been used with up to 25 Gy in 5 fractions (SBRT) without complications [8]. 

The interval between the SFRT fraction(s) and the cERT remains an open question for SFRT in cervical cancer and other primary tumors. An interval of 1–3 days has been proposed and employed successfully in head and neck cancer and sarcoma [5,6,12]. While a 7-day interval would be more advantageous to allow immune activation for potential intra-tumoral immune or abscopal effects postulated in SFRT [32,33], this approach is hampered by the current lack of experience with longer intervals between SFRT and cERT. Furthermore, clinical experience has shown that overall protraction of the treatment course in cervical cancer is associated with a decrease in survival in standard fractionated radiation [34]. Whether this detriment to survival would apply to a lengthening of the treatment course from the addition of SFRT and the associated time interval to cERT remains unknown in cervical cancer and would require careful investigation. 

#### 3.4.6. Brachytherapy 

There is currently minimal experience with the addition of brachytherapy to SFRT and conventional pelvic radiation/concurrent chemotherapy. The only (unpublished) experience to date has employed single-dose LRT of 15 Gy, combined with two fractions of intracavitary/interstitial brachytherapy (according to the EMBRACE regimen) in two highly advanced cervical cancer patients. The panel believes that the experience with combined SFRT, cERT/concurrent chemotherapy and brachytherapy is too sparse at this time to make recommendations on the combination of brachytherapy with SFRT and cERT for a randomized trial in definitively treated cervical cancer. Brachytherapy is an indispensable component of the radiotherapeutic management of cervical cancer and ultimately will have to be incorporated into SFRT regimens for clinical trials. In the absence of solid outcome data, the concerns of added toxicity from the cumulative dose of SFRT, cERT, and brachytherapy remain. The panel therefore recommends that an initial Phase I study be conducted first, which combines SFRT with a conventional cERT regimen (with standard concurrent chemotherapy) and standard brachytherapy. 

For brachytherapy, a more fractionated schedule of at least four fractions, as is commonly used in clinical practice, is recommended, and image guidance for the brachytherapy, preferably with MRI, is strongly encouraged. For the SFRT, a stepwise dose escalation scheme from near-conventional dose (e.g. 4 Gy in 1 fraction, which has already been in use clinically to treat severe tumor hemorrhage [35] to 15 Gy, as is used in the proposed regimen [30], may be considered. A corresponding (biologically isoeffective) multi-fraction regimen, such as Amendola et al.’s [8] three-fraction regimen, is also an option. Such a Phase I trial should contain strict guidelines for normal tissue tolerance doses, commensurate with current recommendations from image-guided brachytherapy. Further consideration should be given to a dosimetric feasibility study (on image-guided ERT and brachytherapy data sets) prior to a Phase I trial in clinical patients. 

#### 3.4.7. Conventional ERT: OAR Constraints

Dose constraints to OARs for the cERT portion of treatment are recommended to follow those in standard practice, such as OAR dose limits established by the RTOG 0921 trial [36]. The contribution of the SFRT dose (converted to EQD2) is included into the determination of the OAR dose for each critical normal structure (high agreement). 

### 3.5. Systemic Therapy

#### 3.5.1. Agents and Timing

Chemotherapy with weekly cisplatin, the standard of care for advanced cervical cancer, should be administered concurrently with radiation therapy. For three-fraction SFRT regimens, chemotherapy has been used concurrently with the first SFRT fraction, based on the study by Amendola et al. that showed minimal toxicity with a regimen of 24 Gy in three consecutive fractions given concurrently with cisplatin chemotherapy. However, in other primary diseases that are treated with concurrent chemoradiation therapy (e.g., head and neck cancer, lung cancer), concurrent chemotherapy has been started 2–3 days after SFRT, concurrently with the start of cERT. Post-radiation adjuvant (“outback”) chemotherapy or adjuvant immunotherapy is not recommended for an initial trial. 

#### 3.5.2. Immunotherapy

There is no published experience with the combination SFRT and immunotherapy in cervical cancer, and the combination of immunotherapy and conventional radiation is currently considered experimental and limited to clinical trials. Therefore, an initial clinical trial of SFRT in cervical cancer should employ standard concurrent weekly cisplatin. Combinations with immunotherapy shall be reserved for future trials. Post-radiation adjuvant immunotherapy, which is also experimental, is not recommended for an initial trial. 

### 3.6. Evaluations and Assessments

Guidelines for patient evaluations are presented in Table 4. Pretreatment clinical, imaging, and histologic investigations are recommended according to standard of care, including clinical examination, inclusive of pelvic exam, blood count and blood chemistries. Chest/abdomen/pelvis CT is the minimum imaging requirement. MRI to define the tumor extent in the pelvis (which can also assist in treatment planning), and PET/CT for identification of lymph node involvement and distant metastases are favored if available (high consensus).

On-treatment evaluations should include standard-of-care weekly toxicity assessments, quality-of-life assessments and patient reported outcomes, and routine imaging that typically includes CBCT imaging for response assessment and adaptive therapy as needed. 

Specimen collection of blood and urine before, multiple times during and after the radiation therapy course for translational correlative science studies of SFRT is feasible and should be strongly considered (high consensus). Such specimens may include investigations of immune status, such as assessments of immune cell phenotypes; circulating cytokines linked to immune activation and radiation sensitivity and circulating tumor cells, markers of senescence, and other investigations [16,17,37,38]. From a biological perspective, collection of such specimen just before SFRT, within a day after SFRT and before cERT, and 7–14 days after SFRT may be most impactful. 

While repeat (cervical) tumor biopsies at the time of brachytherapy may be feasible in selected centers, tumor biopsies once or more than once during therapy, while possible, are overall clinically challenging (moderate consensus). However, collection of pre-therapy tissue is available and should be considered for translational science studies. 

Post-therapy evaluations consist of standard-of-care physical examinations with pelvic exam for response and toxicity assessments (high consensus). Quality-of-life and patient reported outcomes are required. Recommended imaging studies include PET/CT 3 months post-therapy and MRI 1 month post-therapy. Additional imaging is performed if clinically indicated per standard of care (high consensus). 

### 3.7. Knowledge Gaps That May Be Addressed through SFRT Clinical Trials in Gynecologic Cancer 

*Clinical* knowledge gaps centered on a better understanding of clinical feasibility, normal tissue tolerance, best dose and fractionation, and the combination of SFRT with brachytherapy, which could be improved through studies with larger patient numbers. Whether a single SFRT fraction vs. multiple fractions have differing outcomes requires investigation. A better understanding of the best time interval between SFRT and cERT may enhance strategies for engagement of the immune system and rationale for immunotherapy. 

Knowledge gaps in the *physics* and *biology* aspects of SFRT are focused on SFRT technique and dose/fractionation; in particular, whether a single SFRT fraction vs. multiple hypofractions result in different clinical outcomes and/or volume effects. Collection and banking of blood/urine before and after SFRT may provide insights into potential systemic reactions that remain a knowledge gap. 

## 4. Discussion and Conclusions

While the published/presented clinical experience in GRID and Lattice SFRT is relatively sparse, pilot studies defining dosing and techniques in correlation to tumor control and toxicity outcomes have provided a basis for these recommendations to optimally design prospective multi-institutional clinical trials for gynecologic malignancies. Expert clinicians with experience in GRID and more recently LRT-based SFRT have further contributed to these recommendations. 

Among the gynecologic tumors, SFRT clinical trials are feasible in cervical cancer based on the clinical pilot experience in advanced bulky cervical cancer, a gynecologic malignancy where primary radiation remains the mainstay of therapy. Data in other gynecologic cancers, in which radiation therapy plays a major role, are too sparse to justify a clinical trial at this time. 

In cervical cancer, patients with bulky tumors ≥6 cm of squamous cell and/or adenocarcinoma histologies, with or without lymph node involvement, or those unsuitable for brachytherapy, are eligible. LRT is favored over GRID therapy as the SFRT technology based on the currently available clinical experience. While dosing in clinical pilot experience used an LRT regimen of 24 Gy in 3 fractions, a regimen of 15 Gy in 1 LRT fraction may be acceptable; however, outcome data is sparse. As in the SFRT approach for other primary tumors, SFRT must be followed by conventionally fractionated external beam radiation and chemotherapy. 

Standardization of the novel, nonconventional physics of dose inhomogeneity through reporting of dosimetric parameters, particularly EUD, are of paramount importance to allow robust correlation with clinical outcomes. Because the published experience with the combination of SFRT, cERT/concurrent chemotherapy, and standard-of-care brachytherapy is insufficient at this time, a Phase I trial of LRT to test the tolerability of the combined regimen is recommended as an initial study prior to larger-scale trials. Standard-of-care concurrent weekly cisplatin chemotherapy is required. 

Standard pre-, on-, and post-therapy assessments and quality-of-life metrics should be performed. For a prospective SFRT trial, specimen collection (blood, urine), synchronized prospectively with the treatment course, for translational correlative science studies is highly recommended. Uniquely to cervical cancer, intra-therapy tumor biopsy of the cervix can be feasible in selected centers, because cervical tumors are readily accessible clinically, either as a clinic procedure and particularly for biopsies at the time of brachytherapy. This ability to collect tumor tissue, in conjunction with the “liquid biopsy” approaches, may provide unique insights into the response dynamics and biological underpinnings of SFRT that are typically unavailable for many other primary tumors in clinical patients. 

The trial design consensus guideline presented here is based on the current status of knowledge in SFRT for gynecologic cancers. While these recommendations may provide guidance for the design of clinical trials and clinical feasibility considerations for translational studies, the field of SFRT is developing rapidly. Therefore, new data and longer-term outcome results in larger patient series may further refine, adapt, or modify these initial recommendations. The clinical trial design guideline proposed here will have to be individualized by the respective investigators who develop clinical trials in SFRT. 

## Figures and Tables

**Table 1 cancers-14-04267-t001:** Synopsis of Consensus Development Process. Table reprinted with permission from Mayr et al. [28].

Sequence	Process Description
**1. Initial literature** **review**	Search terms: Spatially fractionated radiation therapy, GRID therapy, Lattice therapy, Dose Fractionation, Radiation, Neoplasms/radiotherapy, Neoplasms/pathology, Tumor control Data bases: PubMed, Web of Science, Cochrane Repeat literature search: February 2022
Tabulation of literature into Evidence Table (Appendix A)
**2. Development of initial** **clinical trial design** ** criteria**	Design criteria: Eligibility/exclusion, pre-therapy, on-therapy and post-therapy patient evaluations (for outcome endpoints), endpoints, stratifications, dose and technical radiation therapy factors, clinical feasibility of correlative of studies, concurrent therapies, knowledge gaps that may be addressed in a trial.
Performed by expert group of 3 leaders in general SFRT
**3. Voting Round 1**	Anonymous electronic rating of the appropriateness of the proposed trial design criteria: 20 categories of trial design questions with 2–11 sub-criteria (parameters): Voting scale 1–9 ^1)^ 1 knowledge gap question 1 demographic expertise question
Voters: MD’s, physicists, biologists with clinical experience in SFRT in the disease site and/or publications and/or scientific presentations
**4. Vote analysis and** ** statistical model**	Prioritization of agreement on the broader appropriateness categories, *Appropriate*, *May be Appropriate*, *Not Appropriate* ^1)^ while maintaining the nuancing of the 1–9 scale.
Agreement categories: *High*, *Moderate*, *Low* ^2)^
**5. Review/discussion of** **voting results by** **disease-specific** **Consensus expert** **panel (“Panel”)**	Panel members: 3 radiation oncologists, 2 physicists, 1 biologist with SFRT publications, scientific presentations in the specific disease site, physics or biology, respectively
Consensus development based on voting statistics, literature and the Panel’s clinical/scientific experience
Formal consensus video conference call(s) and consensus communications (email, phone)
**6. Iterative voting ** **round(s)**	Implemented for trial criteria with persistently low agreement, or new trial criteria identified by the panel
**7. Re-review/discussion** **of voting results**	as in step 5 (with or without video conference call)
**8. Draft guideline** **development**	Guideline draft and review by the panel
**9. Public comments**	Public comment posting for 2 weeks (by RSS)
**10. Repeat literature** ** review**	as in step 1
**11. Review/discussion of** ** public comments**	Review of anonymized public comments, as in steps 5 and 7 Guideline revisions as indicated
**12. Final guideline**	Development of final guideline by panel






**^1)^ Voting scale and categories:**
	**Voting scale**
**Voting rank**	**1 2 3**	**4 5 6**	**7 8 9**
**Voting category**	**Not appropriate**for clinical trial design	**May be appropriate**for clinical trial design	**Appropriate**for clinical trial design






**^2)^ Agreement categories of vote—Definitions:**
**Vote Agreement**	**Definition**
**High**	Percent agreement ≥ 67% AND if any disagreement, it is by at most 1 voting category
**Moderate**	60–67% agreement OR agreement ≥67% but votes in both *Appropriate* and *Not appropriate* vote categories
**Low**	Percent agreement <60%

Note for ^1)^: within each voting category, 3 sub-ranks (e.g., 7, 8 and 9) signify ranking as lower, intermediate, and higher appropriateness, respectively. Note for ^2)^: Agreement on the rating of each clinical trial criterion was categorized as either *Low*, *Moderate*, or *High*. *Low agreement* was defined as percent agreement on the broader appropriateness category (*Appropriate*, *May be Appropriate*, and *Not Appropriate*) of less than 60% on the appropriateness category AND no disagreement (if any was present) by more than one category. Thus, ratings of *Appropriate* and *May be Appropriate* or *May be Appropriate* and *Not Appropriate* for the same clinical trial criterion were allowable under *High agreement* if at least two-thirds agreed on a single appropriateness category, while ratings of both *Appropriate* and *Not Appropriate* could not qualify for *High agreement*, regardless of the overall percent agreement. All others were classified as *Moderate agreement*.

**Table 2 cancers-14-04267-t002:** Clinical trial design criteria.

Design Categories	Sub-Categories
**Eligible Disease Sites**	Primary tumor sites
**Eligibility/Exclusion criteria**	Disease stage, tumor size/extent/invasion
	Histology, molecular markers
	Prior treatment
	Patient factors: age, performance status, toxicity risk factors
**Stratifications**	–
**Pre-treatment Evaluations**	Clinical
	Imaging
	Histologic investigations
**Radiation therapy:** **SFRT**	SFRT dose and fractionation
	SFRT target volume
	SFRT OAR constraints
	SFRT technique
**Radiation Therapy:** **Conventional external beam** **radiation therapy**	cERT dose and fractionation
	cERT OAR constraints
	cERT technique
	Brachytherapy
**On-therapy Evaluations**	Clinical
	Laboratory
	Imaging
	Patient-reported outcomes
	Translational (evaluation of clinical feasibility)
**Systemic Therapy**	Cytotoxic agents and timing
	Immunotherapy
**Post-therapy Evaluations**	Clinical
	Imaging
	Patient-reported outcomes
**Knowledge Gaps**	Clinical
	Physics
	Biology/translation science

Note: cERT = conventional external radiation therapy. Table adapted and reprinted with permission from Mayr et al. [28].

**Table 3 cancers-14-04267-t003:** Eligibility, exclusions, and stratifications.

Eligibility Criteria	
**Disease Sites**	Cancer of the cervix
**Stage, tumor size**	Stages IB2–IVA (FIGO_2018_) with tumors ≥6 cm in largest diameter or ≥5 cm and ineligibility for brachytherapy ^1)^ With or without pelvic or paraaortic lymph node involvement
**Histology and tumor markers**	Squamous cell carcinoma, adenocarcinoma, mixed adeno- squamous carcinoma, both HPV-positive and HPV-negative
**Prior therapy**	No prior therapy ^2)^
**Patient factors**	>18 years old No upper age limit if eligible based on performance status (generally ECOG score <2)
**Exclusion criteria**	
**Histology and tumor markers**	Uncommon or highly radiosensitive histologies, such as small cell neuroendocrine carcinoma, sarcoma or lymphoma
**Tumor stage/extent**	Involved supraclavicular lymph nodes or distant metastases
**Prior therapy**	Recurrent tumors after prior radiation therapy Recurrent tumors after prior surgery ^3)^ Prior chemotherapy for cervical cancer
**Patient factors**	Scleroderma (systemic sclerosis) Inflammatory bowel disease
**Stratifications**	
**N-stage grouping**	Lymph node status (uninvolved vs. involved)

Note: ^1)^ ineligibility for brachytherapy because of severe anatomic distortion, such as anatomical or tumor-related severe vaginal atrophy and/or stenosis, obstructing uterine fibroid or other uterine abnormalities, and/or ineligibility for anesthesia from severe (American Society of Anesthesiologists grade IV) medical comorbidities; ^2)^ surgical retroperitoneal, laparoscopic assisted or robotic lymph node dissection (without hysterectomy) prior to radiation therapy is permitted; ^3)^ patients with recurrence after prior hysterectomy are ineligible for a trial of primary cervical cancer (high consensus) but may be considered for a separate trial of recurrent disease if no prior radiation was received.

**Table 4 cancers-14-04267-t004:** Pre-, on-, and post-therapy patient evaluations and assessments.

	Evaluation/Test	Pre-Therapy	On-Therapy	Post-cERT/ Pre-Brachytherapy	During Brachytherapy	Post-Therapy
**Clin**	General physical Exam	**√**	** √** ^1)^	**√**	**√**	** √** ^2)^
	Pelvic Exam	**√**	** √** ^*^	** √** ^3)^	** √** ^4)^	** √** ^2)^
**Imaging**	CT Chest/abd/pelvis CT)	**√**	n/a ^5)^	n/a	n/a	** √** ^*^
	MRI (abdomen/pelvis)	** √** ^6)^	n/a	** √** ^6)^	n/a	** √** ^7)^
	PET/CT	** √** ^8)^	n/a	n/a	n/a	** √** ^9)^
**Lab**	CBC	**√**	**√**	**√**	**√**	**√**
	Blood chemistries	**√**	**√**	**√**	**√**	**√**
**Correl Std**	Blood collection	**√**	** √** ^10)^	**√**	** √** ^*^	** √** ^*^
	Urine collection	**√**	** √** ^11)^	**√**	** √** ^*^	** √** ^*^
	Tumor biopsy/specimen	**√**	** √** ^12)^	n/a	** √** ^13)^	n/a
**PRO**	QOL assessment	**√**	**√**	** √** ^*^	n/a	**√**

Note: √ = recommended; √ * = recommended if clinically indicated; n/a = not recommended. Abd = abdomen; CBC = complete blood count; Clin = clinical; Hist = histology; PRO = patient-reported outcomes; Correl Std = correlative studies. 
^1)^ Weekly.^2)^ Routine follow-up per standard of care, usually every 3 months in year 1–2 and every 4–6 months in year 3–5, yearly thereafter.^3)^ Pelvic exam as performed for response assessment prior to the brachytherapy.^4)^ As indicated during the brachytherapy procedure.^5)^ Cone beam CT can be useful for response assessment and is important for adaptive planning.^6)^ MRI preferred to define the tumor extent in the pelvis, which can also assist in radiation therapy planning pre-treatment and for brachytherapy.^7)^ MRI 1 month post-therapy for response assessment.^8)^ Preferred, if available, for assessment of lymph node involvement and distant metastases.^9)^ PET/CT 3 months post-therapy for response assessment.^10)^ Feasible weekly during radiation therapy along with standard-of-care blood collections for chemotherapy.^11)^ Feasible weekly or at prospective time points/dose levels during/after treatment^12)^ Challenging but potentially feasible.^13)^ Uniquely feasible in cervical cancer during a brachytherapy procedure.

## Data Availability

Study data including the aggregated voting results and the expert panel’s conclusions are available in Appendix A.

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
