# Peer review of "An International Consensus on the Design of Prospective Clinical–Translational Trials in Spatially Fractionated Radiation Therapy for Advanced Gynecologic Cancer"

_cancers, 2022, doi:10.3390/cancers14174267_

Round 1

Reviewer 1 Report

Keywords are missing

Bibliography is incomplete, and also the references to the bibliography in the rest of the article

Tables are missing in the article

Author Response

Response to Reviewer 1 Comments

Manuscript: cancers-1822389

Title: “An International Consensus on the Design of Prospective Clinical–translations Trials in Spatially Fractionated Radiation Therapy for Advanced Gynecologic Cancer

Dear Reviewer, 

Your review and all comments are very much appreciated, and revisions were made accordingly as detailed below.  Edits in the manuscript text are shown in tracking.

Point 1:  Keywords are missing

Response: We thank the Reviewer for identifying this oversight. The following key words were added: “Spatially fractionated radiation therapy; Lattice therapy; GRID therapy; Dose fractionation; Radiation; Clinical trials; Consensus guideline; Gynecologic cancer; Cervix cancer” (page 2, line 66-67 in the tracked version of the revised manuscript). 

Point 2:  Bibliography is incomplete, and also the references to the bibliography in the rest of the article

Response: We included 38 references in the bibliography of the original manuscript, and these are cited in the text of the manuscript.  The bibliography to our knowledge includes all clinical outcome studies of SFRT in gynecologic cancer known to us, and multiple SFRT studies in other malignancies that are relevant to the subject of our work.  Reviewer 2 has noted that our bibliography is a complete review of the literature. 

We believe the bibliography is included and comprehensive, but we will be happy to consider additional references per the Reviewer’s specific recommendation.

Point 3:  Tables are missing in the article

Response: Tables 1-4 have been submitted in the manuscript in the initial submission, and we have located the tables in the submission website (file: cancers-1822389-peer-review.pdf).  They are also included with this revised submission. It is unclear to us how, unfortunately, the tables were not available to the Reviewer, and we hope that the editorial staff may be able to assist if needed. Please let us know if there is anything else we can do to help.

Assessment:  Are the methods adequately described? - must be improved       

Response: The methodology of our consensus is described in section “2 - Materials and Methods” (page 3-6, specifically line 135-171) and further detail is presented in Tables 1 and 2.  The Reviewer’s inability to access Tables 1 and 2 may be related to the assessment of inadequate methods description.

Assessment:  Are the results clearly presented? - must be improved       

Response: Please advise regarding specific points where the presentation of the results should be improved.  Section “3 - SFRT CLINICAL TRIAL DESIGN CONSENSUS GUIDELINE FOR PRIMARY    GYNECOLOGIC CANCER” (page 4-9) are the Results of our study. We did not use the word “Results” in the title of this section, in keeping with standards used for most oncology guidelines publications in the literature.  Please kindly advise on the specific issues.  We have, however, added the term “Results” to the header of section 3 for clarification (page 4, line 173).  

Please find uploaded the edited manuscript with changes in tracking. 

Thank you again for your review.  Please feel free to contact us if we may provide any further information.

Sincerely,

Nina A. Mayr, MD, FASTRO, FAAAS, FAAWR

Professor and Senior Academic Advisor to the Executive Vice President for Health Sciences, Michigan State University, East Lansing, MI, 48824

Reviewer 2 Report

Dear Authors, 

This article focuses on an important topic. Since there is little data in the literature and considering the authors' experience in this field, this study is essential in identifying the guidelines that will underlie the best experimental and clinical practices. The authors provided adequate details on methodology, evaluation, findings, and investigations. The particularities and novelty of the article are very well underlined in the results and conclusions sections. Given the bibliography, it is clear that the authors made a complete review of the literature beforehand. Overall, this manuscript is very well written and documented, representing an important moment in deciphering the treatment strategies for gynecologic cancers.

However, some suggestions could improve the quality of the article:

-       Line 256 et al. instead of and al

-       Line 291 typo mistake computation computation

-       The lack of randomized clinical trials in managing these difficult clinical cases for conventional RT makes it impossible to optimize the SFRT clinical regimen or correlate the peak dose regarding the SFRT dosimetry with the tumor response or tissue toxicity.

-       SFRT dosimetry is important in highlighting cellular cytotoxicity and triggering processes related to the infiltration of immune cells or vascular permeability.

-       To be highlighted in the studies related to the clinical versus the preclinical SFRT of the differences in the size of the peak and the width of the valley, as well as the ratio of the peak dose to the valley.

-       A possible combination of SFRT and Ultrahigh dose-rate radiotherapy (FLASH-RT) could represent a future strategy.

-       Emphasize conventional imaging (cone beam computed tomography) and magnetic resonance or positron emission tomography imaging techniques to monitor effects for linear accelerator or FLASH particles.

-       The dosimetric study of SFRT using high-quality plans at the cervix level, respecting the constraints for the neighboring organs at risk, can reach the dose and the effects during brachytherapy.

-       The rapid planning technique of SFRT can be an option for neoadjuvant treatment to be adopted in radio-oncology clinics, allowing a dose escalation to deep masses to eliminate large unresectable tumors.

Kind regards

Author Response

Response to Reviewer 2 Comments

Manuscript: cancers-1822389

Title: “An International Consensus on the Design of Prospective Clinical–translations Trials in Spatially Fractionated Radiation Therapy for Advanced Gynecologic Cancer

Dear Reviewer, 

Your review and all comments are very much appreciated, and revisions were made accordingly as detailed below.  Edits in the manuscript text are shown in tracking.

Thank you for your positive comments regarding relevance of this subject, our methodology, evaluation, investigations and findings

Point 1:  Line 256 et al. instead of and al  

Response: Thank you for identifying this typo. It was corrected (page 4, line 230 of the revised manuscript).

Point 2:  Line 291 typo mistake computation computation

Thank you. We corrected the removed the duplication (page 6, line 268).

Point 3:  The lack of randomized clinical trials in managing these difficult clinical cases for conventional RT makes it impossible to optimize the SFRT clinical regimen or correlate the peak dose regarding the SFRT dosimetry with the tumor response or tissue toxicity.

Response: We agree that there is (despite the recently reported Outback Trial in stage IB-IVA cervical cancer) a lack of clinical trials in far-advanced/bulky cervical cancer as a separate group.  We are more optimistic and believe that this lack of randomized trials may not make it impossible to optimize the SFRT regimen.  A viable SFRT regimen has already been reported (Amendola et al. (1), and this can be further refined. We believe that the greater obstacle to further development and study of SFRT in cervical cancer is the lack of data on combining brachytherapy with SFRT, and we have pointed this out in the manuscript (page 6, line 251-253). 

Point 4:  SFRT dosimetry is important in highlighting cellular cytotoxicity and triggering processes related to the infiltration of immune cells or vascular permeability.

Response: We agree completely.  The importance of dosimetry has been stated in the manuscript (page 6, line 254-259).  We have now added a statement that robust dosimetry is important for biological effects on the tumor microenvironment and potential immune modulation in SFRT, as follows: “Robust dosimetry and dosimetry reporting are also important for advancing the understanding of tumor environmental and potential immune modulation effects of SFRT” (page 6, line 259-261). 

Point 5:  To be highlighted in the studies related to the clinical versus the preclinical SFRT of the differences in the size of the peak and the width of the valley, as well as the ratio of the peak dose to the valley.

Response: Thank you for pointing out the differences between clinical and preclinical dose heterogeneity patterns. A statement regarding this was added to the Introduction: “In addition, dose heterogeneity patterns, which differ from those in preclinical SFRT studies, are variable.” (page 3, line 104-105).   

Point 6:  A possible combination of SFRT and Ultrahigh dose-rate radiotherapy (FLASH-RT) could represent a future strategy. 

Response: While this may be possible in the future, our current clinical trial design consensus solely focuses on GRID and Lattice technologies that are ready to be tested in a human trial.  Although SFRT–FLASH combination may be a promising option in the future, no clinical facilities exist to deliver this option at this time.  The Consensus Panel believes that it would be beyond the scope (and perhaps confusing to the reader) to discuss a combination with FLASH within this clinical trial design guideline for GRID- and Lattice-based SFRT in cervical cancer.  However, if the Reviewer feels strongly, we would consider adding a statement in the manuscript on the current status on FLASH/SFRT combination and add reference (3). 

Point 7:  Emphasize conventional imaging (cone beam computed tomography) and magnetic resonance or positron emission tomography imaging techniques to monitor effects for linear accelerator or FLASH particles.

Response: We agree with the importance of both conventional (CT-based), MRI and PET imaging for the assessment of treatment effects and response in patients treated with SFRT in gynecologic cancer.  Fortunately, all these imaging techniques are now more widely used in cervical cancer.  Imaging for response assessment has already been described in Table 4 (table: rows 5 and 6, and foot notes 6 and 8). We also mentioned the importance of using cone beam computed tomography (CBCT) in the original manuscript (line 323: “CBCT imaging is particularly important due to the frequently observed rapid tumor response that may require adaptive therapy”).  We now added CBCT to Table 4, as follows: “Cone beam CT can be useful for response assessment and is important for adaptive planning” (Table 4, row 4 and footnote 5).   Thank you for pointing out this omission in Table 4.

Because our consensus is about the design of clinical trials with GRID or Lattice based SFRT, which are ready to be used the humans in the clinical environment, and because and there is no pilot literature on the SFRT–FLASH combination, our Consensus Panel believes that for clinical trials of SFRT in cervical cancer, cone beam CT, MRI and PET should be used as morphologic and molecular imaging response assessment.  However, tracking of response to FLASH beam particles may not be feasible in a current clinical trial until FLASH therapy becomes better understood and more widely applicable in radiation oncology centers for gynecologic cancer.  

Point 8:  The dosimetric study of SFRT using high-quality plans at the cervix level, respecting the constraints for the neighboring organs at risk, can reach the dose and the effects during brachytherapy. 

Response: Because there is to date no published peer-reviewed study on the use of SFRT instead of brachytherapy, our Consensus Panel considers this approach experimental at this time.  Because of the importance of brachytherapy for overall treatment in cervical cancer and the generally low toxicity of SFRT (1), the Consensus Panel believes that there is a good chance that SFRT can be successfully combined with brachytherapy. We recommend that this combination be tested first in a pilot study, as stated on page 10, line 447-450 (“Because the published experience with the combination of SFRT, cERT/concurrent chemotherapy and standard-of-care brachytherapy is insufficient at this time, a Phase I trial of LRT to test the tolerability of the combined regimen is recommended”).  

Point 9:  The rapid planning technique of SFRT can be an option for neoadjuvant treatment to be adopted in radio-oncology clinics, allowing a dose escalation to deep masses to eliminate large unresectable tumors.

Response: We agree that this is possible, and the cervical cancer series (1) and the case report (2) Amendola et al. support the Reviewer’s assessment by showing high complete response rates in bulky unresectable cervical cancer.  We also agree that rapid planning and re-planning are needed, and this has now been emphasized in the manuscript. We also added a note on rapid replanning, as follows: “CBCT imaging is particularly important due to the frequently observed rapid tumor response that may require adaptive therapy using rapid replanning.” (page 7, line 301-302, added text printed in italics). 

Please find uploaded the edited manuscript with changes in tracking. 

Thank you again for your review.  Please feel free to contact us if we may provide any further information.

Sincerely,

Nina A. Mayr, MD, FASTRO, FAAAS, FAAWR

Professor and Senior Academic Advisor to the Executive Vice President for Health Sciences, Michigan State University, East Lansing, MI, 48824

References:

  1. Amendola, B.E.; Perez, N.C.; Mayr, N.A.; Wu, X.; Amendola, M. Spatially Fractionated Radiation Therapy Using Lattice Radiation in Far-advanced Bulky Cervical Cancer: A Clinical and Molecular Imaging and Outcome Study. Radiat Res 2020, 194, 724-736, doi:10.1667/RADE-20-00038.1.

  1. Amendola, B.E.; Perez, N.; Amendola, M.; X., W.; Ahmed, M.M.; Iglesias, A.J.; Estape, R.; Lambrou, N.; Bortoletto, P. Lattice radiotherapy with RapidArc for treatment of gynecological tumors: dosimetric and early clinical evaluations. Cureus 2010, 2, 1-6.

  1. Schneider, T.; Fernandez-Palomo, C.; Bertho, A.; et al. Combining FLASH and spatially fractionated radiation therapy: the best of both worlds. Radiother Oncol 2022, S0167-8140(22)04226-8. 8 Aug. 2022, doi:10.1016/j.radonc.2022.08.004

Round 2

Reviewer 1 Report

Corrections has been realized correctly